# Node Classification on Graphs with Few-Shot Novel Labels via Meta Transformed Network Embedding

**Lin Lan**[1]*, **Pinghui Wang**[1]*†, **Xuefeng Du**[2]*, **Kaikai Song**[3], **Jing Tao**[1], **Xiaohong Guan**[1,4]

[1] MOE Key Laboratory of Intelligent Networks and Network Security,
School of Automation Science and Engineering, Xi'an Jiaotong University, China
[2] University of Wisconsin-Madison, USA    [3] Huawei Noah's Ark Lab, China
[4] Department of Automation and NLIST Lab, Tsinghua University, China
llan@sei.xjtu.edu.cn, {phwang, jtao, xhguan}@mail.xjtu.edu.cn
xdu66@wisc.edu, caesar.song@huawei.com

## Abstract

We study the problem of node classification on graphs with few-shot novel labels, which has two distinctive properties: (1) There are novel labels to emerge in the graph; (2) The novel labels have only a few representative nodes for training a classifier. The study of this problem is instructive and corresponds to many applications such as recommendations for newly formed groups with only a few users in online social networks. To cope with this problem, we propose a novel Meta Transformed Network Embedding framework (MetaTNE), which consists of three modules: (1) A *structural module* provides each node a latent representation according to the graph structure. (2) A *meta-learning module* captures the relationships between the graph structure and the node labels as prior knowledge in a meta-learning manner. Additionally, we introduce an *embedding transformation function* that remedies the deficiency of the straightforward use of meta-learning. Inherently, the meta-learned prior knowledge can be used to facilitate the learning of few-shot novel labels. (3) An *optimization module* employs a simple yet effective scheduling strategy to train the above two modules with a balance between graph structure learning and meta-learning. Experiments on four real-world datasets show that MetaTNE brings a huge improvement over the state-of-the-art methods.

## 1 Introduction

Graphs are ubiquitously used to represent data in a wide range of fields, including social network analysis, bioinformatics, recommender systems, and computer network security. Accordingly, graph analysis tasks, such as node classification, link prediction, and community detection, have a significant impact on our lives in reality. In this paper, we focus on the task of node classification. Particularly, we consider the classification of *few-shot novel labels*, which means there are some novel labels to emerge in the graph of interest and the novel labels usually have only a few representative nodes including the positive and the negative (i.e., holding and not holding the novel labels, respectively). Hereafter, we refer to the *available positive and negative nodes* of a label as the *support nodes* of that label. The study of *Node Classification on graphs with Few-shot Novel Labels* (NCFNL) is instructive for many practical applications. Let us consider the following scenarios.

**Motivating Examples**. (1) Some organizations in online social networks, such as Facebook, Twitter, and Flickr, may distribute advertisements about whether users are interested in their new features or

are willing to join their new social media groups. Through NCFNL, these organizations can predict other users' preferences based on positive and negative responses of a few users and provide better services or recommendations without too much bother for users. (2) For biological protein-protein networks, some researchers may discover a new biological function of certain proteins. Given a few proteins with and without a specific function, the study of NCFNL could predict whether other proteins have the function, which helps recommend new directions for wet laboratory experimentation.

Some straightforward ways could be derived from existing unsupervised or semi-supervised network embedding methods while suffer from low performance, and please refer to § 2 for detailed discussions. To tackle this problem, we argue that different labels in a graph share some intrinsic evolution patterns (e.g., the way a label propagates along the graph structure according to the proximities between nodes). Assuming that there are a set of labels that have sufficient support nodes (e.g., interest groups that have existed and evolved for a long time in online social networks and protein functions that biologists are already familiar with), we desire to extract the common patterns from the graph structure and these labels and then utilize the found patterns to help recognize few-shot novel labels. However, the relationships between the graph structure and node labels are complex and there could be various propagation patterns between nodes. It remains challenging to design a model to capture all the patterns, and how to apply them to novel labels still needs to be further studied.

**Overview of Our Approach.** Inspired by recent advances in few-shot learning through meta-learning [22, 7, 25, 13], we cast the problem of NCFNL as a meta-learning problem and propose a novel Meta Transformed Network Embedding framework, namely MetaTNE, which allows us to exploit the common patterns. As shown in Fig. 1, our proposed framework consists of three modules: *the structural module*, *the meta-learning module*, and *the optimization module*. Given a graph and a set of labels (called known labels) with sufficient support nodes, the structural module first learns a latent representation for each node according to the graph structure. Then, considering that we ultimately expect to recognize few-shot novel labels, we propose the meta-learning module to simulate the few-shot scenario during the training phase instead of directly performing optimization over all known labels. Moreover, most existing meta-learning works [7, 25] focus on image- and text-related tasks, while the graph structure is more irregular in nature. To adequately exploit the complex and multifaceted relationships between nodes, we further design an *embedding transformation function* to map the structure-only (or task-agnostic) node representations to the task-specific ones for different few-shot classification tasks. To some extent, the meta-learning module implicitly encodes the shared propagation patterns of different labels through learning a variety of tasks. Finally, the optimization module is proposed to train the preceding two modules with a simple yet effective scheduling strategy in order to ensure the training stability and the effectiveness. One advantage of MetaTNE is that, after training, it is natural to directly apply the learned meta-learning module to few-shot novel labels.

Our main contributions are summarized as follows:

- We explore to only use the graph structure and some known labels to study the problem of NCFNL. Compared with previous graph convolution based works [40, 34] that rely on high-quality node content for feature propagation and aggregation, our work is more challenging and at the same time more applicable to content-less scenarios.

- We propose an effective framework to solve NCFNL in a meta-learning manner. Our framework is able to generalize to classifying emerging novel labels with only a few support nodes. In particular, we design a transformation function that captures the multifaceted relationships between nodes to facilitate applying meta-learning to the graph data.

- We conduct extensive experiments on four publicly available real-world datasets, and empirical results show that MetaTNE achieves up to 150.93% and 47.58% performance improvement over the state-of-the-art methods in terms of Recall and $F_1$, respectively.

## 2 Related Work

**Unsupervised Network Embedding.** This line of works focus on learning node embeddings that preserve various structural relations between nodes [37, 2], including skip-gram based methods [20, 26, 8, 23], deep learning based methods [4, 30], and matrix factorization based methods [3, 21]. A straightforward way to adapt these methods for NCFNL is to simply train a new classifier (e.g., logistic regression) when novel labels emerge, while the learned node embeddings hold constant. However,

this does not incorporate the guidance from node labels into the process of network embedding, which dramatically degrades the performance in the few-shot setting.

**Semi-Supervised Network Embedding.** These approaches typically formulate a unified objective function to jointly optimize the learning of node embeddings and the classification of nodes, such as combining the objective functions of DeepWalk and support vector machines [15, 27], as well as regarding labels as a kind of context and using node embeddings to simultaneously predict structural neighbors and node labels [6, 32]. Another line of works [11, 9, 29, 10, 31] explore graph neural networks to solve semi-supervised node classification as well as graph classification. Two recent works [16, 39] extend graph convolutional network (GCN) [11] to accommodate to the few-shot setting. However, the above methods are limited to a fixed set of labels and the adaptation of them to NCFNL requires to train the corresponding classification models or parameters from scratch when a novel label appears, which is not a well-designed solution to the few-shot novel labels and usually cannot reach satisfactory performance. Recently, Chauhan et al. [5] study few-shot graph classification with unseen novel labels based on graph neural networks. Zhang et al. [36] propose a few-shot knowledge graph completion method that essentially performs link prediction in a novel graph given a few training links. In comparison, we study node classification with respect to few-shot novel labels in the same graph and their methods are not applicable.

In addition, GCN based methods **heavily rely on** high-quality node content for feature propagation and aggregation, while in some networks (e.g., online social networks), some nodes (e.g., users) may not expose or expose noisy (low-quality) content, or even all node content is unavailable due to privacy concerns [38, 14], which would limit the practical use of these methods. In contrast, our focus is to solve the problem of NCFNL by exploiting the relationships between the graph structure and the node labels, without involving node content.

**Meta-Learning on Graphs.** Zhou et al. [40] propose Meta-GNN that applies MAML [7] to GCN in a meta-learning way. More recently, Yao et al. [34] propose a method that combines GCN with metric-based meta-learning [25]. To some extent, all methods could handle novel labels emerging in a graph. However, they are built upon GCN and thus need high-quality node content for better performance, while in this paper we are interested in graphs without node content.

**Few-Shot Learning on Images.** Recently, few-shot learning has received considerable attention. Most works [22, 7, 25, 33, 18] focus on the problem of few-shot image classification in which there are no explicit relations between images. Some works also introduce task-specific designs for better generalization and learnability, such as task-specific null-space projection [35] and infinite mixture prototypes [1]. However, graph-structured data exhibits complex relations between nodes (i.e., the graph structure) which are the most fundamental and important information in a graph, making it difficult to directly apply these few-shot methods to graphs. In addition, Liu et al. [17] propose to construct a graph of image classes and learn to propagate messages between prototypes of different classes according to the graph structure, of which the goal is to obtain better class prototypes for few-shot image classification. Although this work introduces the concept of graph meta-learning, it is not applicable to our scenario where a label can be positive or negative for different nodes.

## 3  Problem Formulation

Throughout the paper, we use lowercase letters to denote scalars (e.g., $\ell$), boldface lowercase letters to denote vectors (e.g., $\mathbf{u}$), and boldface uppercase letters to denote matrices (e.g., $\mathbf{W}$).

We denote a graph of interest by $\mathcal{G} = (\mathcal{V}, \mathcal{E}, \mathcal{Y})$, where $\mathcal{V} = \{v_1, v_2, \ldots, v_{|\mathcal{V}|}\}$ is the set of nodes, $\mathcal{E} = \{e_{ij} = (v_i, v_j)\} \subseteq \mathcal{V} \times \mathcal{V}$ is the set of edges, and $\mathcal{Y}$ is the set of labels associated with nodes in the graph. Here, we consider the multi-label setting where each node may have multiple labels. Let $\ell_{v_i, y} \in \{0, 1\}$ be the label indicator of the node $v_i$ in terms of the label $y \in \mathcal{Y}$, where $\ell_{v_i, y} = 1$ suggests that the node $v_i$ holds the label $y$ and $\ell_{v_i, y} = 0$ otherwise. We use $\mathcal{D}_y^+ = \{v_i | \ell_{v_i, y} = 1\}$ to denote nodes that hold the label $y$, and $\mathcal{D}_y^- = \{v_i | \ell_{v_i, y} = 0\}$ to denote nodes that do not hold the label $y$. In this paper, we assume $\mathcal{G}$ is undirected for ease of presentation.

**Known Labels and Novel Labels.** We divide the labels into two categories: the known labels $\mathcal{Y}_{\text{known}}$ and the novel labels $\mathcal{Y}_{\text{novel}}$. The former are given before we start any kind of learning process (e.g., semi-supervised network embedding), while the latter emerge after we have learned a model.

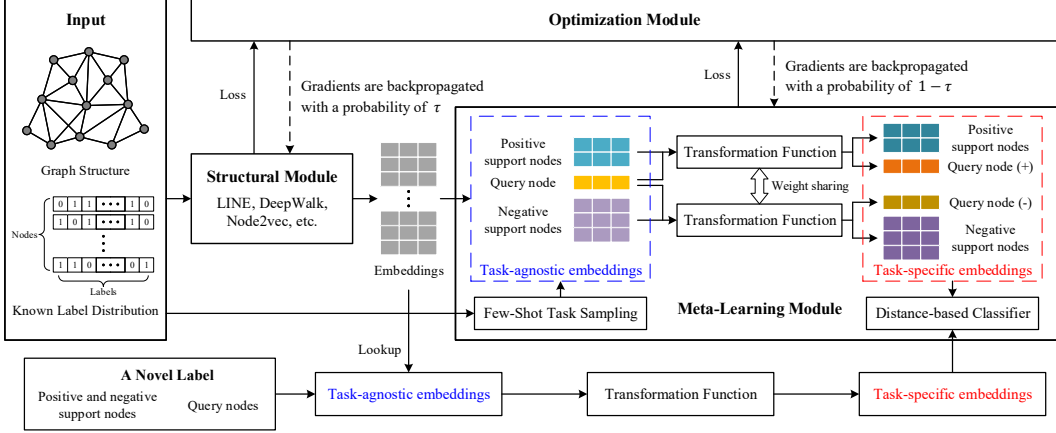

Figure 1: A schematic depiction of our MetaTNE. In the meta-learning module, we use 2 positive and 3 negative support nodes for simplicity of illustration. The threshold $\tau$ gradually decreases from 1 to 0 during training. The flow of applying MetaTNE to a novel label is shown at the bottom.

We assume that each known label is complete, namely $|\mathcal{D}_y^+| + |\mathcal{D}_y^-| = |\mathcal{V}|$ for $y \in \mathcal{Y}_{\text{known}}$. To some extent, the known labels refer to relatively stable labels (e.g., an interest group that has existed and evolved for a long time in online social networks). Although for some nodes, inevitably we are not sure whether they hold specific known labels or not, we simply assume that the corresponding label indicators equal 0 (i.e., not holding) like many other node classification works [20, 26]. In practice, a more principled way is to additionally consider the case of uncertain node-label pairs and define the label indicator as 1, 0, and -1 for the cases of holding the label, uncertain label, and not holding the label, respectively, which we leave as future work.

On the other hand, a novel label has only a few support nodes (e.g., 10 positive nodes and 10 negative nodes). By leveraging the known labels that have sufficiently many positive and negative nodes, we aim to explore the propagation patterns of labels along the graph structure and learn a model that generalizes well to classifying emerging novel labels with only a few support nodes.

**Our Problem.** Given a graph $\mathcal{G} = (\mathcal{V}, \mathcal{E}, \mathcal{Y}_{\text{known}}, \mathcal{Y}_{\text{novel}})$, the problem of NCFNL aims to explore the relationships between the graph structure and the known labels $\mathcal{Y}_{\text{known}}$ and learn a generalizable model for classifying novel labels $\mathcal{Y}_{\text{novel}}$. Specifically, for each $y \in \mathcal{Y}_{\text{novel}}$, after observing only a few corresponding support nodes, the model should be able to generate or act as a good classifier to determine whether other nodes hold the label $y$ or not.

## 4 Algorithm

In this section, we present our proposed MetaTNE in detail, which consists of three modules: the structural module, the meta-learning module, and the optimization module, as shown in Fig. 1. Given a graph and some known labels, the structural module learns an embedding for each node based on the graph structure. Then, the meta-learning module learns a transformation function that adapts the structure-only node embeddings for each few-shot node classification task sampled from the known labels and performs few-shot classification using a distance-based classifier. Finally, to optimize our model, we propose a learning schedule that optimizes the structural and meta-learning modules with probabilities that gradually decrease and increase from 1 to 0 and from 0 to 1, respectively.

### 4.1 Structural Module

The structural module aims to learn a representation or embedding in the latent space for each node while preserving the graph structure (i.e., the connections between nodes). Mathematically, for each node $v_i \in \mathcal{V}$, we maximize the log-probability of observing its neighbors by optimizing the following objective function: $\min \sum_{v_i \in \mathcal{V}} \sum_{v_j \in \mathcal{N}(v_i)} \log \mathbb{P}(v_j | v_i)$, where $\mathcal{N}(v_i)$ denotes the neighboring nodes of $v_i$. We optimize the above objective function following the skip-gram architecture [19]. Regarding the construction of the neighboring set $\mathcal{N}(\cdot)$, although there are many choices such as 1-hop neighbors

based on the connectivity of nodes [26] and the random walk strategy [20, 8], in this paper we adopt the 1-hop neighbors for the sake of simplicity. By optimizing the above objective, we are able to obtain an embedding matrix $\mathbf{U} \in \mathbb{R}^{|V| \times d}$, of which the $i$-th row $\mathbf{u}_i$ indicates the representation of $v_i$.

## 4.2 Meta-Learning Module

As alluded before, we cast the problem of NCFNL into the meta-learning framework [22, 7] and simulate the few-shot setting with $\mathcal{Y}_{\text{known}}$ during training. In what follows, we first describe how to organize the graph structure and the known labels in the meta-learning scenario. Then, we give a metric-based meta-learning paradigm for solving NCFNL. In particular, we propose a transformation function that transforms the task-agnostic embeddings to the task-specific ones in order to better deal with the multi-label setting where each node may be associated with multiple labels.

### 4.2.1 Data Organization

Instead of directly optimizing over the entire set of known labels like traditional semi-supervised learning methods [32], we propose to construct a pool of few-shot node classification tasks according to the known labels $\mathcal{Y}_{\text{known}}$. Analogous to few-shot image classification tasks in the literature of meta-learning [22], a few-shot node classification task $\mathcal{T}_i = (\mathcal{S}_i, \mathcal{Q}_i, y_i)$ is composed of a support set $\mathcal{S}_i$, a query set $\mathcal{Q}_i$, and a label identifier $y_i$ randomly sampled from $\mathcal{Y}_{\text{known}}$. The support set $\mathcal{S}_i = \mathcal{S}_i^+ \cup \mathcal{S}_i^-$ contains the set $\mathcal{S}_i^+$ of randomly sampled positive nodes and the set $\mathcal{S}_i^-$ of randomly sampled negative nodes, where $\mathcal{S}_i^+ \subset \mathcal{D}_{y_i}^+$ and $\mathcal{S}_i^- \subset \mathcal{D}_{y_i}^-$. The query set $\mathcal{Q}_i = \mathcal{Q}_i^+ \cup \mathcal{Q}_i^-$ is defined in the same way but does not intersect with the support set, namely $\mathcal{Q}_i^+ \subset \mathcal{D}_{y_i}^+ \setminus \mathcal{S}_i^+$ and $\mathcal{Q}_i^- \subset \mathcal{D}_{y_i}^- \setminus \mathcal{S}_i^-$. The task is, given the support set of node-label pairs, finding a classifier $f_{\mathcal{T}_i}$ which is able to predict the probability $\hat{\ell}_{v_q, y_i} \in [0, 1]$ for each query node $v_q$ with a low misclassification rate. We denote by $\mathcal{T}_i \sim p(\mathcal{T}|\mathcal{Y}_{\text{known}})$ sampling a few-shot node classification task from $\mathcal{Y}_{\text{known}}$.

### 4.2.2 Meta-Learning with Embedding Transformation for NCFNL

To facilitate learning to classify for a label with few associated nodes in a graph, we apply a meta-learning flavored learning scheme. Following the above definition of few-shot node classification tasks, for each task $\mathcal{T}_i = (\mathcal{S}_i, \mathcal{Q}_i, y_i) \sim p(\mathcal{T}|\mathcal{Y}_{\text{known}})$, we aim to construct a classifier $f_{\mathcal{T}_i}$ for the label $y_i$ given the support set $\mathcal{S}_i$, which is able to classify the query nodes in the set $\mathcal{Q}_i$. Formally, for each $(v_q, \ell_{v_q, y_i}) \in \mathcal{Q}_i$, the classification loss is defined as follows:

$$\mathcal{L}(\hat{\ell}_{v_q, y_i}, \ell_{v_q, y_i}) = -\ell_{v_q, y_i} \log \hat{\ell}_{v_q, y_i} - (1 - \ell_{v_q, y_i}) \log(1 - \hat{\ell}_{v_q, y_i}), \tag{1}$$

where $\hat{\ell}_{v_q, y_i}$ denotes the predicted probability that $v_q$ holds label $y_i$. Here, to calculate the probability, we adopt a distance-based classifier which is commonly used in the metric-based meta-learning literature [25]. Specifically, for each task $\mathcal{T}_i$, the classifier $f_{\mathcal{T}_i}$ is parametrized by two $d$-dimensional latent representations, $\mathbf{c}_+^{(i)}$ (called positive prototype) and $\mathbf{c}_-^{(i)}$ (called negative prototype), that correspond to the cases of holding and not holding label $y_i$, respectively. The predictions are made based on the distances between the node representations and these two prototypes. Mathematically, given the embedding vector $\mathbf{u}_q$ of each query node $v_q$, we have the predicted probability as

$$\hat{\ell}_{v_q, y_i} = f_{\mathcal{T}_i}(v_q | \mathbf{c}_+^{(i)}, \mathbf{c}_-^{(i)}) = \frac{\exp(-\text{dist}(\mathbf{u}_q, \mathbf{c}_+^{(i)}))}{\sum_{m \in \{+, -\}} \exp(-\text{dist}(\mathbf{u}_q, \mathbf{c}_m^{(i)}))}, \tag{2}$$

where $\text{dist}(\cdot, \cdot) : \mathbb{R}^d \times \mathbb{R}^d \to [0, +\infty)$ is the squared Euclidean distance function and the positive or negative prototype is usually calculated as the mean vector of node representations in the corresponding support set [25].

**Why do we need Embedding Transformation?** Equation (2) makes predictions under the condition that each node is represented by the same or *task-agnostic* embedding vector regardless of which label or task we are concerned about. Technically, this scheme makes sense for few-shot image classification in prior works [25] where each image is assigned to the same one and only one label. However, this is problematic in the multi-label scenario where each node could be assigned to multiple labels. Here is an illustrating example. In social networks, suppose we have two classification tasks $\mathcal{T}_1$ and $\mathcal{T}_2$ with respect to different labels, namely "Sports" from $\mathcal{Y}_{\text{known}}$ and "Music" from $\mathcal{Y}_{\text{novel}}$,

and two users $A$ and $B$ are involved in these two tasks. Both users $A$ and $B$ could give positive feedback to "Sports", while on the other hand, they could give positive and negative feedback to "Music" respectively. Intuitively, the task-agnostic scheme may provide similar embeddings after fitting well on the task $\mathcal{T}_1$, which is not appropriate for the task $\mathcal{T}_2$.

**High-Level Module Design.** To mitigate the above problem, we propose to learn a transformation function $Tr(\cdot)$ which transforms the task-agnostic embeddings to some task-specific ones for each task. First, we argue that different query nodes have different correlation patterns with the nodes in the support set. To fully explore how a query node correlates with the support nodes, we propose to tailor the embeddings of the support nodes for each query node. Second, to classify a query node, we are more interested in characterizing the distance relationship between the query node and either positive or negative support nodes rather than the relationship between the positive and negative support nodes. Thus, during the transformation, we propose to adapt the query node with the positive and the negative nodes in the support set separately.

Based on the above two principles, for each query node, we first construct two sets: one containing the task-agnostic embeddings of the query node and the positive support nodes, and the other containing the task-agnostic embeddings of the query node and the negative support nodes. Then, we separately feed the two sets into the transformation function. The meta-learning module in Fig. 1 illustrates this process. Formally, given a task $\mathcal{T}_i = (\mathcal{S}_i, \mathcal{Q}_i, y_i)$, for each query node $v_q \in \mathcal{V}_{\mathcal{Q}_i}$, we have

$$
\begin{aligned}
\{\tilde{\mathbf{u}}_{q,+}^{(i)}\} \cup \{\tilde{\mathbf{u}}_{k,q}^{(i)}|v_k \in \mathcal{V}_{\mathcal{S}_i^+}\} = Tr(\{\mathbf{u}_q\} \cup \{\mathbf{u}_k|v_k \in \mathcal{V}_{\mathcal{S}_i^+}\}), \\
\{\tilde{\mathbf{u}}_{q,-}^{(i)}\} \cup \{\tilde{\mathbf{u}}_{k,q}^{(i)}|v_k \in \mathcal{V}_{\mathcal{S}_i^-}\} = Tr(\{\mathbf{u}_q\} \cup \{\mathbf{u}_k|v_k \in \mathcal{V}_{\mathcal{S}_i^-}\}),
\end{aligned}
\tag{3}
$$

where $\tilde{\mathbf{u}}_{q,+}^{(i)}$ and $\tilde{\mathbf{u}}_{q,-}^{(i)}$ denote the adapted embedding of the query node $v_q$ in relation to the positive and negative support nodes, respectively, and $\tilde{\mathbf{u}}_{k,q}^{(i)}$ denotes the adapted embedding of the support node $v_k$ tailored for the query node $v_q$. As a result, each query node has two different adapted embeddings $\tilde{\mathbf{u}}_{q,+}^{(i)}$ and $\tilde{\mathbf{u}}_{q,-}^{(i)}$ that are further used for comparisons with the adapted embeddings of the positive and negative support nodes, respectively. A consequential benefit is that the transformation function is more flexible to capture the multifaceted relationships between nodes in the multi-label scenario. Imagine that even if the task-specific embeddings of the positive and negative support nodes or prototypes are distributed close, we are still able to make right predictions through altering $\tilde{\mathbf{u}}_{q,+}^{(i)}$ and $\tilde{\mathbf{u}}_{q,-}^{(i)}$. The ablation study in Section Experiments and the visualization in the supplement confirm the superiority of this design.

**Instantiation.** As per the above discussions, we propose to implement $Tr(\cdot)$ using the self-attention architecture with the scaled dot-product attention mechanism [28], which has exhibited the ability to effectively capture relationships between a set of elements. We start with some basic concepts of the self-attention. Each input element plays three different roles in the self-attention: (1) It is compared with every other element to compute the weights that indicate *how much it attends to other elements*; (2) It is compared with every other element to compute the weights that indicate *how much other elements attend to it*; (3) It is used *as part of the output of each element* after the weights between elements have been determined. Following the prior work [28], for each input element, we establish three vectors, the *query* vector, the *key* vector, and the *value* vector, to represent the three roles, respectively. Typically, these three vectors are obtained by applying linear transformations to the input vector of each element with three trainable matrices, which enables us to learn to make each element suit the three roles it needs to play.

Next, we elaborate on how to leverage the self-attention architecture to instantiate Eqn. (3), which separately takes as input the two sets $\{\mathbf{u}_q\} \cup \{\mathbf{u}_k|v_k \in \mathcal{V}_{\mathcal{S}_i^m}\}$ where $m \in \{+,-\}$. For any two nodes $v_i, v_j \in \{v_q\} \cup \mathcal{V}_{\mathcal{S}_i^m}$ ($v_i$ and $v_j$ could be the same), we first calculate the attention $\omega_{ij}$ that $v_i$ pays to $v_j$ as follows:

$$
\omega_{ij} = \frac{\exp((\mathbf{W}_Q\mathbf{u}_i) \cdot (\mathbf{W}_K\mathbf{u}_j)/d'^{1/2})}{\sum_{v_k \in \{v_q\} \cup \mathcal{V}_{\mathcal{S}_i^m}} \exp((\mathbf{W}_Q\mathbf{u}_i) \cdot (\mathbf{W}_K\mathbf{u}_k)/d'^{1/2})},
\tag{4}
$$

where $\mathbf{W}_Q, \mathbf{W}_K \in \mathbb{R}^{d' \times d}$ denote the trainable matrices that project the input vectors into the *query* and *key* vectors, respectively, $d'$ denotes the dimension of the *query*, *key*, and *value* vectors, "$\cdot$" denotes the dot product operator, and $\frac{1}{\sqrt{d'}}$ is a scaling factor to avoid extremely small gradients [28].

In effect, the attention $\omega_{ij}$ reflects the degree to which the node $v_j$ relates to or influences $v_i$. Then, the output or transformed vector of each node aggregates information from every other node in a weighted manner. Specifically, let $\mathbf{W}_V \in \mathbb{R}^{d' \times d}$ be the trainable matrix to calculate the *value* vectors and $\mathbf{W}_O \in \mathbb{R}^{d \times d'}$ be another trainable matrix to ensure that the output vectors are of the same dimension as the input vectors. We compute the output vector of the query node $v_q$ as

$$\tilde{\mathbf{u}}_{q,m}^{(i)} = \mathbf{W}_O \left( \omega_{qq} \mathbf{W}_V \mathbf{u}_q + \sum_{v_k \in \mathcal{V}_{\mathcal{S}_i^m}} \omega_{qk} \mathbf{W}_V \mathbf{u}_k \right), \tag{5}$$

and compute the output vector of each support node $v_k \in \mathcal{V}_{\mathcal{S}_i^m}$ tailored for the query node $v_q$ as

$$\tilde{\mathbf{u}}_{k,q}^{(i)} = \mathbf{W}_O \left( \omega_{kk} \mathbf{W}_V \mathbf{u}_k + \sum_{v_j \in \left( \mathcal{V}_{\mathcal{S}_i^m} \backslash \{v_k\} \right) \cup \{v_q\}} \omega_{kj} \mathbf{W}_V \mathbf{u}_j \right). \tag{6}$$

We refer readers to the supplement for more details on the instantiation of the transformation function.

With the transformed embeddings, we further calculate the positive and negative prototypes tailored for $v_q$ as well as the predicted probability as follows:

$$\tilde{\mathbf{c}}_{m,q}^{(i)} = \frac{1}{|\mathcal{S}_i^m|} \sum_{v_k \in \mathcal{V}_{\mathcal{S}_i^m}} \tilde{\mathbf{u}}_{k,q}^{(i)}, \ m \in \{+,-\}, \text{ and } \hat{\ell}_{v_q,y_i} = \frac{\exp(-\mathrm{dist}(\tilde{\mathbf{u}}_{q,+}^{(i)}, \tilde{\mathbf{c}}_{+,q}^{(i)}))}{\sum_{m \in \{+,-\}} \exp(-\mathrm{dist}(\tilde{\mathbf{u}}_{q,m}^{(i)}, \tilde{\mathbf{c}}_{m,q}^{(i)}))}. \tag{7}$$

The final meta-learning objective is formulated as:

$$\min_{\mathbf{U},\Theta} \sum_{\mathcal{T}_i} \sum_{(v_q, \ell_{v_q,y_i}) \in \mathcal{Q}_i} \mathcal{L}(\hat{\ell}_{v_q,y_i}, \ell_{v_q,y_i}) + \lambda \sum \|\Theta\|_2^2, \tag{8}$$

where $\mathcal{T}_i \sim p(\mathcal{T}|\mathcal{Y}_{\mathrm{known}})$, $\hat{\ell}_{v_q,y_i}$ is calculated through Eqn. (7), $\Theta$ refers to the set of parameter matrices (e.g., $\mathbf{W}_Q$, $\mathbf{W}_K$, and $\mathbf{W}_V$) contained in $Tr(\cdot)$, and $\lambda > 0$ is a balancing factor.

### 4.3 Optimization and Using the Learned Model for Few-Shot Novel Labels

For optimization, one typical way is to minimize the (weighted) sum of the structural loss and the meta loss. However, the structure information of the graph is still not properly embedded at the beginning of the training stage, and the node representations are somewhat random which make no sense for the few-shot classification tasks. Therefore, a training procedure that focuses on optimizing the structural module at the beginning and then gradually pays more attention to optimizing the meta-learning module is preferably required. To satisfy this requirement, we take inspiration from learning rate annealing [12] and introduce a probability threshold $\tau$, and in each training step the structural and meta modules are optimized with probabilities of $\tau$ and $1 - \tau$, respectively. The probability threshold $\tau$ is gradually decayed from 1 to 0 in a staircase manner, namely $\tau = 1/(1 + \gamma \lfloor \frac{step}{N_{\mathrm{decay}}} \rfloor)$ where $\gamma$ is the decay rate, $step$ is the current step number, and $N_{\mathrm{decay}}$ indicates how often the threshold is decayed. The complete optimization procedure is outlined in the supplement. In addition, the time complexity is analyzed in the supplement.

Recall that our ultimate goal is to, after observing a few support nodes associated with a novel label $y \in \mathcal{Y}_{\mathrm{novel}}$, predict whether other (or some query) nodes have the label $y$ or not. In effect, this can be regarded as a few-shot node classification task $\mathcal{T} = (\mathcal{S}, \mathcal{Q}, y)$. After optimization, we have obtained the task-agnostic node representations $\mathbf{U}$, and the transformation function $Tr(\cdot)$ parameterized by $\Theta$. Thus, to classify a query node $v_q \in \mathcal{Q}$, we simply look up the representations of the query node and the support nodes from $\mathbf{U}$, adapt their representations using the transformation function as formulated in Eqn. (5) and (6), and compute the predicted probability according to Eqn. (7). The detailed procedure is presented in the supplement.

Table 1: Statistics of the datasets.

| Dataset | #Nodes | #Edges | #Labels |
|---|---|---|---|
| BlogCatalog | 10,312 | 333,983 | 39 |
| Flickr | 80,513 | 5,899,882 | 195 |
| PPI | 3,890 | 76,584 | 50 |
| Mashup | 16,143 | 300,181 | 28 |

Table 2: Results on few-shot node classification tasks with novel labels. OOM means out of memory (16 GB GPU memory). The standard deviation is provided in the supplement.

(a) $K_{*,+} = 10$ and $K_{*,-} = 20$.

| Method | BlogCatalog | | | Flickr | | | PPI | | | Mashup | | |
|---|---|---|---|---|---|---|---|---|---|---|---|---|
| | AUC | $F_1$ | Recall | AUC | $F_1$ | Recall | AUC | $F_1$ | Recall | AUC | $F_1$ | Recall |
| LP | 0.6422 | 0.1798 | 0.2630 | 0.8196 | 0.4321 | 0.4989 | 0.6285 | 0.2147 | 0.2769 | 0.6488 | 0.3103 | 0.4535 |
| LINE | 0.6690 | 0.2334 | 0.1595 | 0.8593 | 0.6194 | 0.5418 | 0.6372 | 0.2147 | 0.1456 | 0.6926 | 0.2970 | 0.2142 |
| Node2vec | 0.6697 | 0.3750 | 0.2940 | 0.8504 | 0.6664 | 0.6147 | 0.6273 | 0.3545 | 0.2860 | 0.6575 | 0.3835 | 0.3147 |
| Planetoid | 0.6850 | 0.4657 | 0.4301 | **0.8601** | 0.6638 | 0.6331 | 0.6791 | 0.4672 | 0.4411 | 0.7056 | 0.4825 | 0.4218 |
| GCN | 0.6643 | 0.3892 | 0.3379 | OOM | OOM | OOM | 0.6596 | 0.4176 | 0.3729 | 0.6910 | 0.4065 | 0.3607 |
| Meta-GNN | 0.6533 | 0.3567 | 0.2962 | OOM | OOM | OOM | 0.6537 | 0.3964 | 0.3373 | 0.7093 | 0.4689 | 0.4202 |
| MetaTNE | **0.6986** | **0.5380** | **0.6203** | 0.8462 | **0.7118** | **0.7700** | **0.6865** | **0.5188** | **0.5621** | **0.7645** | **0.5764** | **0.5566** |
| %Improv. | 1.99 | 15.53 | 44.22 | -1.62 | 6.81 | 21.62 | 1.09 | 11.04 | 27.43 | 7.78 | 19.46 | 22.73 |

(b) $K_{*,+} = 10$ and $K_{*,-} = 40$.

| Method | BlogCatalog | | | Flickr | | | PPI | | | Mashup | | |
|---|---|---|---|---|---|---|---|---|---|---|---|---|
| | AUC | $F_1$ | Recall | AUC | $F_1$ | Recall | AUC | $F_1$ | Recall | AUC | $F_1$ | Recall |
| LP | 0.6421 | 0.0554 | 0.0727 | 0.8253 | 0.3055 | 0.3040 | 0.6298 | 0.0773 | 0.0748 | 0.6534 | 0.1156 | 0.1284 |
| LINE | 0.6793 | 0.0529 | 0.0328 | 0.8644 | 0.4154 | 0.3485 | 0.6423 | 0.0496 | 0.0300 | 0.7009 | 0.0956 | 0.0617 |
| Node2vec | 0.6792 | 0.1982 | 0.1340 | 0.8558 | 0.5295 | 0.4602 | 0.6309 | 0.1894 | 0.1306 | 0.6643 | 0.2070 | 0.1447 |
| Planetoid | 0.6981 | 0.2980 | 0.2319 | **0.8728** | 0.5040 | 0.4461 | 0.6879 | 0.3100 | 0.2523 | 0.7095 | 0.3279 | 0.2551 |
| GCN | 0.6794 | 0.2104 | 0.1583 | OOM | OOM | OOM | 0.6608 | 0.2531 | 0.1974 | 0.7007 | 0.2558 | 0.2098 |
| Meta-GNN | 0.6724 | 0.2152 | 0.1618 | OOM | OOM | OOM | 0.6617 | 0.2575 | 0.2088 | 0.7140 | 0.3412 | 0.2864 |
| MetaTNE | **0.7139** | **0.4398** | **0.5819** | 0.8505 | **0.6220** | **0.7460** | **0.7039** | **0.4298** | **0.5327** | **0.7684** | **0.4814** | **0.4816** |
| %Improv. | 2.26 | 47.58 | 150.93 | -2.55 | 17.47 | 62.10 | 2.33 | 38.65 | 111.14 | 7.62 | 41.09 | 68.16 |

# 5 Experiments

Four publicly available real-world benchmark datasets are used to validate the effectiveness of our method. The statistics of these datasets are summarized in Table 1. For each dataset, we split the labels into training, validation, and test labels according to a ratio of 6:2:2. In the training stage, we regard the training labels as the known labels and sample few-shot node classification tasks from them. For validation and test purposes, we regard the validation and test labels as the novel labels and sample 1,000 tasks from them, respectively. We use the average classification performance on the test tasks for comparisons of different methods. For ease of presentation, we use $K_{\mathcal{S},+}$, $K_{\mathcal{S},-}$, $K_{\mathcal{Q},+}$, and $K_{\mathcal{Q},-}$ to indicate the respective numbers of positive support, negative support, positive query, and negative query nodes in a task. We compare MetaTNE with Label Propagation [41], unsupervised network embedding methods (LINE [26] and Node2vec [8]), semi-supervised network embedding methods (Planetoid [32] and GCN [11]), and Meta-GNN [40]. For detailed experimental settings including dataset and baseline descriptions, baseline evaluation procedure, and parameter settings, please refer to the supplement.

**Overall Comparisons.** Following the standard evaluation protocol of meta-learning [7], we first compare different methods with $K_{\mathcal{S},+} = K_{\mathcal{Q},+}$ and $K_{\mathcal{S},-} = K_{\mathcal{Q},-}$ (hereafter using $K_{*,+}$ and $K_{*,-}$ for simplicity), and these numbers are the same for both training and test tasks. Considering that negative samples are usually easier than positive samples to acquire we report the overall performance with $K_{*,+}$ set to 10 and $K_{*,-}$ set to 20 and 40, respectively. The comparison results on the four datasets are presented in Table 2. Since in our application scenarios we prefer to discover proteins with new functions in biological networks and find users who are interested in the latest advertisements on online social networks rather than predict negative samples accurately, we report Recall in addition to AUC and $F_1$. To eliminate randomness, all of the results here and in the following quantitative experiments are averaged over 50 different trials.

From Table 2, we observe that MetaTNE consistently and significantly outperforms all other methods in terms of the three metrics across all the four datasets except the AUC scores on Flickr dataset. By jointly analyzing the $F_1$ and Recall scores, MetaTNE predicts positive nodes from imbalanced data more effectively than the baselines, with little loss of precision. In particular, MetaTNE achieves 44.22% and 150.93% gains over the strongest baseline (i.e., Planetoid) with respect to Recall on BlogCatalog dataset when $K_{*,-}$ equals 20 and 40, respectively.

Compared with the unsupervised methods, Planetoid reaches better performance owing to the use of training labels. On the other hand, GCN also uses training labels as supervision, while does not show satisfactory performance and even worse performance than Node2vec, which is due to that the graph convolution relies heavily on node attributes for feature propagation and aggregation as mentioned before and the lack of node attributes limits its representativeness and thus classification capacity.

Besides, Meta-GNN underperforms the unsupervised methods and GCN in some cases, which seems to contradict the published results in the original paper. The reasons are twofold: (1) Meta-GNN is built upon GCN and the predictive ability is also limited due to the lack of node attributes, while the original paper focuses on attributed graphs; (2) Meta-GNN simply applies MAML to GCN and is originally used for the multi-class setting (e.g., each document has the same and only one label in Cora [24]). However, we consider the multi-label setting and the same pair of nodes may have opposite relations in different tasks, which will introduce noisy and contradictory signals in the optimization process of MAML and further degrade the performance in some cases.

**Ablation Study.** In what follows, to gain deeper insight into the contributions of different components involved in our approach, we conduct ablation studies by considering the following variants: (1) a variant without the transformation function; (2) a variant that produces task-specific embeddings by simply feeding all support and query node representations into the self-attention network instead of according to Eqn. (3); (3) a variant that optimizes the total loss of the two modules with the meta-learning loss scaled by a balancing factor searched over

Table 3: Results of ablation study in terms of $F_1$.

| Method | $K_{*,+} = 10, K_{*,-} = 20$ | | $K_{*,+} = 10, K_{*,-} = 40$ | |
|---|---|---|---|---|
| | BlogCatalog | PPI | BlogCatalog | PPI |
| MetaTNE | **0.5380** | **0.5188** | **0.4398** | **0.4298** |
| V1 | 0.5028 | 0.4851 | 0.3998 | 0.3721 |
| V2 | 0.5020 | 0.5011 | 0.4141 | 0.4078 |
| V3 | 0.5205 | 0.4980 | 0.4039 | 0.4074 |
| V4 | 0.4748 | 0.4614 | 0.3549 | 0.3389 |
| V5 | 0.4892 | 0.4819 | 0.3699 | 0.3777 |

$\{10^{-2}, 10^{-1}, \cdots, 10^2\}$; (4) the node embeddings are learned at the beginning and then left fixed (i.e., the structural and meta-learning losses are optimized separately); (5) each node is represented by a one-hot vector and the node embeddings are only optimized with respect to the meta-learning loss. We refer to these variants as V1, V2, V3, V4, and V5. The results are summarized in Table 3.

We see that MetaTNE consistently outperforms its ablated variants. Especially, comparing MetaTNE with V1, we confirm the necessity to introduce the transformation function. The comparison with V2 demonstrates the effectiveness of our special design in Eqn (3). The results of V3 and V4 indicate that our proposed scheduling strategy can boost the performance of MetaTNE with a better balance between the two modules during optimization. The results of V5 show that it is important to introduce the structural loss to optimize the node embeddings. In addition, we see that V1 underperforms V2 even if the node embeddings of V1 are first learned from the graph structure. We speculate that the reason is that at the beginning, the latent space of node embeddings somewhat overfits to the metric of graph structure learning, making it harder to adapt to the metric of subsequent meta-learning or few-shot learning tasks.

**Additional Experiments.** In the supplement, we present more analytical experiments on the numbers of support and query nodes, and illustrate the effect of the proposed transformation function through a visualization experiment.

# 6 Conclusion and Future Work

This paper studies the problem of node classification on graphs with few-shot novel labels. To address this problem, we propose a new semi-supervised framework MetaTNE that integrates network embedding and meta-learning. Benefiting from utilizing known labels in a meta-learning manner, MetaTNE is able to automatically capture the relationships between the graph structure and the node labels as prior knowledge and make use of the prior knowledge to help recognize novel labels with only a few support nodes. Extensive experiments on four real-world datasets demonstrate the superiority of our proposed method. In the future, to improve the interpretability, we plan to extend our approach to quantify the relationships between different labels (e.g., the weight that one label contributes to another) during meta-learning. Another interesting idea is to explicitly incorporate the graph structure information into the meta-learning module, such as developing a more principled way to construct few-shot tasks according to the graph structure instead of random sampling.

## Broader Impact

In general, this work has potential positive impact on graph-related fields that need to deal with the classification problem with respect to few-shot novel labels. For instance, our work is beneficial for social networking service providers such as Facebook and Twitter. These providers can obtain quick and effective feedback on newly developed features through distributing surveys among a small group of users on social networks. In addition, our work can also help biologists, after discovering a new function of certain existing proteins, quickly understand whether other proteins in a protein-protein interaction network have the new function, which improves the efficiency of wet laboratory experimentation. Moreover, many recommender systems model users and items as a graph and enhance the recommendation performance with the aid of network embedding. To some extent, our work is potentially useful to alleviate the cold-start problem as well.

At the same time, our model could be biased towards the few-shot setting after training and not provide superior performance on those labels with many support nodes. In practice, if the original few-shot label gradually has enough support nodes (e.g., biologists identify more proteins with and without the new function through laboratory experiments), we recommend using general unsupervised or semi-supervised methods (e.g., Node2vec [8] or Planetoid [32]) to recognize the label.

## Acknowledgments and Disclosure of Funding

The research presented in this paper is supported in part by National Natural Science Foundation of China (61922067, U1736205, 61902305), MoE-CMCC "Artifical Intelligence" Project (MCM20190701), Natural Science Basic Research Plan in Shaanxi Province of China (2019JM-159), Natural Science Basic Research Plan in Zhejiang Province of China (LGG18F020016).

## Footnotes

*Lin, Pinghui, and Xuefeng contributed equally to this work.

†Pinghui Wang is the corresponding author.

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
