[Supplementary Material]

# Supplement to Node Classification on Graphs with Few-Shot Novel Labels via Meta Transformed Network Embedding

## 1 Additional Algorithm Details

### 1.1 Details of the Transformation Function

For the transformation function, we stack multiple computation blocks as shown in Fig. 1. The stacking mechanism helps the function capture comprehensive relationships between nodes such that the performance is boosted. In each computation block, there are mainly two modules. The first is a self-attention module used to capture the relationships between input nodes, and the second is a node-wise fully-connected feed-forward network used to introduce nonlinearity. In addition, following [11], we employ a residual connection around each of the self-attention module and the feed-forward network and then perform layer normalization, in order to make the optimization faster and more stable.

Figure 1: Illustration of the transformation function. The support nodes are either positive or negative.

The detailed architecture of the self-attention module is illustrated in Fig. 2. Following [11], we extend the self-attention with multiple parallel attention heads using multiple sets of trainable matrices (i.e., $\mathbf{W}_Q^h, \mathbf{W}_K^h, \mathbf{W}_V^h \in \mathbb{R}^{\frac{d'}{H} \times d}$ where $h = 1, \ldots, H$). In each attention head (i.e., each scaled dot-product attention), for any two nodes $v_i, v_j \in \{v_q\} \cup \mathcal{V}_{\mathcal{S}_i^m}$ ($v_i$ and $v_j$ could be the same and $m \in \{+, -\}$) within task $\mathcal{T}_i$, we first calculate the attention $\omega_{ij}$ that $v_i$ pays to $v_j$ as follows:

$$\omega_{ij}^h = \frac{\exp((\mathbf{W}_Q^h \mathbf{u}_i) \cdot (\mathbf{W}_K^h \mathbf{u}_j)/\sqrt{d'/H})}{\sum_{v_k \in \{v_q\} \cup \mathcal{V}_{\mathcal{S}_i^m}} \exp((\mathbf{W}_Q^h \mathbf{u}_i) \cdot (\mathbf{W}_K^h \mathbf{u}_k)/\sqrt{d'/H})}, \quad (1)$$

where "$\cdot$" denotes the dot product operator. Then, we compute the output vector of the query node $v_q$ as

$$\tilde{\mathbf{u}}_{q,m}^{i,h} = \omega_{qq}^h \mathbf{W}_V^h \mathbf{u}_q + \sum_{v_k \in \mathcal{V}_{\mathcal{S}_i^m}} \omega_{qk}^h \mathbf{W}_V^h \mathbf{u}_k, \quad (2)$$

and compute the output vector of each support node $v_k \in \mathcal{V}_{\mathcal{S}_i^m}$ tailored for the query node $v_q$ as

$$\tilde{\mathbf{u}}_{k,q}^{i,h} = \omega_{kk}^h \mathbf{W}_V^h \mathbf{u}_k + \sum_{v_j \in (\mathcal{V}_{\mathcal{S}_i^m} \setminus \{v_k\}) \cup \{v_q\}} \omega_{kj}^h \mathbf{W}_V^h \mathbf{u}_j. \quad (3)$$

Finally, we concatenate the output vectors of all attention heads and use a trainable matrix $\mathbf{W}_O \in \mathbb{R}^{d \times d'}$ to project the concatenated vectors into the original space with the input dimension:

$$\tilde{\mathbf{u}}_{q,m}^{(i)} = \mathbf{W}_O(\tilde{\mathbf{u}}_{q,m}^{i,1} \oplus \cdots \oplus \tilde{\mathbf{u}}_{q,m}^{i,H}), \quad \text{and} \quad \tilde{\mathbf{u}}_{k,q}^{(i)} = \mathbf{W}_O(\tilde{\mathbf{u}}_{k,q}^{i,1} \oplus \cdots \oplus \tilde{\mathbf{u}}_{k,q}^{i,H}), \forall v_k \in \mathcal{V}_{\mathcal{S}_i^m}. \quad (4)$$

Figure 2: Illustration of the self-attention module. The support nodes are either positive or negative.

The multiple parallel attention heads allow the function to jointly attend to information from different input nodes for each input node, and thus help the function better exploit the relationships between input nodes.

## 1.2 Pseudo Codes

The optimization procedure is outlined in Algorithm 1. The procedure of using the learned model for few-shot novel labels is presented in Algorithm 2.

## 1.3 Time Complexity Analysis

For the structural module, we optimize the objective function in a way similar to [9] and the time complexity is $O(kd|\mathcal{E}|)$ where $k$ is the number of negative nodes at each iteration, $d$ is the dimension of node embeddings, and $|\mathcal{E}|$ is the number of edges. For the meta-learning module, the time cost mainly comes from the embedding transformation through the self-attention architecture [11]. Specifically, let $m$ be the number of query nodes and $n$ be the number of positive or negative support nodes. Calculating the *query*, *key*, and *value* vectors takes $O(mndd')$, where $d'$ is the dimension of the *query*, *key*, and *value* vectors. Calculating the attention weights and the weighted sum of *value* vectors takes $O(mn^2d')$. Calculating the final output vectors takes $O(mndd')$. Overall, the time complexity of MetaTNE is $O(kd|\mathcal{E}| + mndd' + mn^2d')$. Note that we can take advantage of GPU acceleration for optimization in practice.

# 2 Details of the Experimental Settings

## 2.1 Datasets

Four datasets are used in our experiments.

**BlogCatalog** [10]: This dataset is the friendship network crawled from the BlogCatalog website. The friendships and group memberships are encoded in the edges and labels, respectively.[1]

**Flickr** [10]: This dataset is the friendship network among the bloggers crawled from the Flickr website. The friendships and group memberships are encoded in the edges and the labels, respectively.[2]

**PPI** [2]: This dataset is a protein-protein interaction network for Homo Sapiens. Different labels represent different function annotations of proteins.[3]

**Mashup** [13]: This dataset is a protein-protein interaction network for human. Different labels represent different function annotations of proteins.[4]

**Algorithm 1** The Optimization Procedure of MetaTNE

---

**Input:** Graph $G$, total number of steps $N$, decay rate $\gamma$, decay period $N_{\text{decay}}$
**Output:** The embedding matrix $\mathbf{U} \in \mathbb{R}^{|V| \times M}$, the function $Tr(\cdot)$
 1: Randomly initialize $\mathbf{U}$ and the parameters $\Theta$ of $Tr(\cdot)$
 2: **for** $step = 0$ **to** $N$ **do**
 3:     Calculate the threshold $\tau = 1/(1 + \gamma \left\lfloor \frac{step}{N_{\text{decay}}} \right\rfloor)$
 4:     Draw a random number $r \sim \text{Uniform}(0,1)$
 5:     **if** $r < \tau$ **then** $\qquad\qquad\qquad\qquad\qquad\qquad\qquad$ ▷ Optimize the structural module
 6:         Sample a batch of pairs $\{(v_i, v_j)|v_i \in \mathcal{V}, v_j \in \mathcal{N}(v_i)\}$
 7:         Update $\mathbf{U}$ to optimize the objective function:

$$\min \sum_{v_i \in \mathcal{V}} \sum_{v_j \in \mathcal{N}(v_i)} \log \mathbb{P}(v_j|v_i) \tag{5}$$

 8:     **else** $\qquad\qquad\qquad\qquad\qquad\qquad\qquad\qquad\qquad$ ▷ Optimize the meta-learning module
 9:         Sample a batch of tasks $\mathcal{T}_i$ from $\mathcal{Y}_{known}$
10:         **for all** $\mathcal{T}_i = (\mathcal{S}_i, \mathcal{Q}_i, y_i)$ **do**
11:             **for all** $v_q \in \mathcal{Q}_i$ **do**
12:                 Calculate the adapted embeddings $\{\tilde{\mathbf{u}}_{k,q}^{(i)}|v_k \in \mathcal{V}_{\mathcal{S}_i^m}\}$ and $\tilde{\mathbf{u}}_{q,m}^{(i)}$, where $m \in \{+,-\}$, via Eqn. (4)
13:                 Calculate the prototypes $\tilde{\mathbf{c}}_{+,q}^{(i)}$ and $\tilde{\mathbf{c}}_{-,q}^{(i)}$:

$$\tilde{\mathbf{c}}_{m,q}^{(i)} = \frac{1}{|\mathcal{S}_i^m|} \sum_{v_k \in \mathcal{V}_{\mathcal{S}_i^m}} \tilde{\mathbf{u}}_{k,q}^{(i)}, \ m \in \{+,-\} \tag{6}$$

14:                 Calculate the predicted probability that $v_q$ holds $y_i$:

$$\hat{\ell}_{v_q,y_i} = \frac{\exp(-\text{dist}(\tilde{\mathbf{u}}_{q,+}^{(i)}, \tilde{\mathbf{c}}_{+,q}^{(i)}))}{\sum_{m \in \{+,-\}} \exp(-\text{dist}(\tilde{\mathbf{u}}_{q,m}^{(i)}, \tilde{\mathbf{c}}_{m,q}^{(i)}))} \tag{7}$$

15:             **end for**
16:         **end for**
17:         Update $\mathbf{U}$ and $\Theta$ to optimize the objective function:

$$\min_{\mathbf{U},\Theta} \sum_{\mathcal{T}_i} \sum_{(v_q, \ell_{v_q,y_i}) \in \mathcal{Q}_i} \mathcal{L}(\hat{\ell}_{v_q,y_i}, \ell_{v_q,y_i}) + \lambda \sum \|\Theta\|_2^2, \tag{8}$$

18:     **end if**
19: **end for**

---

---

**Algorithm 2** Applying MetaTNE to Few-Shot Novel Labels

---

**Input:** The embedding matrix $\mathbf{U}$, the function $Tr(\cdot)$, a novel label $y \in \mathcal{Y}_{novel}$, associated positive support nodes $\mathcal{V}_{\mathcal{S}+}$ and negative support nodes $\mathcal{V}_{\mathcal{S}-}$, query nodes $\mathcal{V}_{\mathcal{Q}}$
**Output:** The predicted probability $\hat{\ell}_{v_q,y}$ for each query node $v_q$
 1: Look up in $\mathbf{U}$ to get the support and query embeddings $\mathbf{u}_k, \mathbf{u}_q$.
 2: **for** $v_q$ in $\mathcal{V}_{\mathcal{Q}}$ **do**
 3:     Adapt $v_q$ together with $\mathcal{V}_{\mathcal{S}+}$ according to Eqn. (4) and obtain adapted embeddings $\{\tilde{\mathbf{u}}_{q,+}\} \cup \{\tilde{\mathbf{u}}_{k,q}|v_k \in \mathcal{V}_{\mathcal{S}+}\}$.
 4:     Adapt $v_q$ together with $\mathcal{V}_{\mathcal{S}-}$ according to Eqn. (4) and obtain adapted embeddings $\{\tilde{\mathbf{u}}_{q,-}\} \cup \{\tilde{\mathbf{u}}_{k,q}|v_k \in \mathcal{V}_{\mathcal{S}-}\}$.
 5:     Calculate the positive and negative prototypes $\tilde{\mathbf{c}}_{m,q}, m \in \{+,-\}$ for classification according to Eqn. (6).
 6:     Calculate the predicted probability with $\tilde{\mathbf{c}}_{m,q}$ and $\tilde{\mathbf{u}}_{q,m}$ according to Eqn. (7).
 7: **end for**

---

## 2.2 Baselines

The following baselines are considered:

**Label Propagation (LP)** [15]: This method is a semi-supervised learning algorithm that estimates labels by propagating label information through a graph. It assigns a node the label which most of its neighborhoods have and propagates until no label is changing.

**LINE** [9]: This method first separately learns node embeddings by preserving 1- and 2-step neighborhood information between nodes and then concatenates them as the final node embeddings.

**Node2Vec** [2]: This method converts graph structure to node sequences by mixing breadth- and depth-first random walk strategies and learns node embeddings with the skip-gram model [8].

**GCN** [4]: This method is a semi-supervised method that uses a localized first-order approximation of spectral graph convolutions to exploit the graph structure. Here we use the learned Node2Vec embeddings as the input feature matrix of GCN.

**Planetoid** [12]: This is a semi-supervised method that learns node embeddings by using them to jointly predict node labels and node neighborhoods in the graph.

**Meta-GNN** [14]: This method directly applies MAML [1] to train GCN [4] in a meta-learning manner. Similarly, we use the learned Node2Vec embeddings as the input feature matrix of GCN.

**Baseline Evaluation Procedure.** We assess the performance of the baselines on the node classification tasks sampled from the test labels as follows: (1) For LP, we propagate the labels of the support nodes over the entire graph and inspect the predicted labels of the query nodes for each test tasks; (2) For each unsupervised network embedding method, we take the learned node embeddings as features to train a logistic regression classifier with L2 regularization for each test task. We use the support set to train the classifier and then predict the labels of the query nodes; (3) For each semi-supervised network embedding method, we first use the training labels to train the model for multi-label node classification. Then, for each test task, we fine-tune the model by substituting the final classification layer with a binary classification layer. Analogous to (2), we use the support set to train the new layer and then predict the labels of the query nodes; (4) For Meta-GNN, we first employ MAML [1] to learn a good initialization of GCN on the training tasks (binary node classification tasks). Then, for each test task, we use the support set to update the GCN from the learned initialization and apply the adapted GCN to the query nodes.

## 2.3 Parameter Settings

For LP, we use an open-source implementation[5] and set the maximum iteration number to 30. For fair comparisons, we set the dimension of node representations to 128 for LINE, Node2vec, and Planetoid. For LINE, we set the initial learning rate to 0.025 and the number of negative samples to 5. For Node2vec, we set the window size to 10, the length of each walk to 40, and the number of walks per node to 80. The best in-out and return hyperparameters are tuned on the validation tasks with a grid search over $p, q \in \{0.25, 0.5, 1, 2, 4\}$. For Planetoid, we use the variant Planetoid-G since there are no input node features in our datasets. We tune the respective batch sizes and learning rates used for optimizing the supervised and the structural objectives based on the performance on the validation tasks. For GCN, we use a two-layer GCN with the number of hidden units as 128 and ReLU nonlinearity, and tune the dropout rate, learning rate, and weight decay based on the performance on the validation tasks. and set other hyperparameters as the original paper. For Meta-GNN[6], we also use a two-layer GCN with 128 hidden units and ReLU nonlinearity. We set the number of inner updates to 2 due to the limitation of GPU memory and tune the fast and meta learning rates based on the performance on the validation tasks. For Planetoid, GCN, and Meta-GNN, we apply the best performing models on the validation tasks to the test tasks.

For our proposed MetaTNE, there are three parts of hyperparameters. In the structural module, we need to set the size $d$ of node representations and sample $N_1$ node pairs at each training step. We also sample $N_{\text{neg}}$ negative nodes per pair to speed up the calculation as in [9]. In the meta-learning module, we sample $N_2$ training tasks at each training step. The hyperparameters involved in the

Table 1: The hyperparameter search space.

| Hyperparameter | Values | Hyperparameter | Values |
|---|---|---|---|
| $N_1$ | $\{512, 1024, 2048\}$ | $L$ | $\{1, 2, 3\}$ |
| $N_2$ | $\{32, 64, 128\}$ | $\lambda$ | $\{0.001, 0.01, 0.1\}$ |
| $H$ | $\{1, 2, 4\}$ | $\alpha_1$ | $\{0.0001, 0.001\}$ |
| $d'$ | $\{128, 256\}$ | $\alpha_2$ | $\{0.0001, 0.001\}$ |
| $d_{\text{ff}}$ | $\{256, 512\}$ | $N_{\text{decay}}$ | $\{500, 1000, 1500, 2000\}$ |

transformation function include the number $H$ of parallel attention heads, the size $d'/H$ of the query, key, and value vectors, the size $d_{\text{ff}}$ of the hidden layer in the two-layer feed-forward network, the number $L$ of stacked computation blocks. Besides, we apply dropout to the output of each of the self-attention modules and the feed-forward networks before it is added to the corresponding input and normalized, and the dropout rate is denoted by $P_{\text{drop}}$. Another hyperparameter is the weight decay coefficient $\lambda$. In the optimization module, we use the Adam optimizer [3] to optimize the structural and the meta-learning modules with learning rates of $\alpha_1$ and $\alpha_2$, respectively. In addition, we have the decay rate $\gamma$ and the decay period $N_{\text{decay}}$ to control the optimization of the structural and meta-learning modules.

For all four datasets, we set $d = 128$, $N_{\text{neg}} = 5$, $P_{\text{drop}} = 0.1$, and $\gamma = 0.1$. We tune other hyperparameters on the validation tasks over the search space shown in Table 1. We utilize the Ray Tune library [6] with asynchronous HyperBand scheduler [5] to accelerate the searching process. Note that, for each dataset, we only search the best hyperparameters with $K_{*,+} = 10$ and $K_{*,-} = 20$ for both training and test tasks, and directly apply these hyperparameters to other experimental scenarios. The resulting hyperparameters are available in our attached code.

## 3 Additional Experiments

### 3.1 Full Results of Overall Comparisons

The full results of overall comparisons in our original paper are presented in Table 2 in the form of $\text{mean} \pm \text{std}$. Overall, we observe that our proposed MetaTNE achieves comparable or even lower standard deviation, which demonstrates the statistical significance of the superiority of MetaTNE.

### 3.2 The Performance w.r.t. the Numbers of Positive and Negative Nodes

To further investigate the performance under different combinations of $K_{*,+}$ and $K_{*,-}$, we conduct experiments with $K_{*,+}$ fixed at either 10 or 20 while varying $K_{*,-}$ from 10 to 50 for both training and test tasks. Figure 3 gives the performance comparisons of MetaTNE and the best performing baseline (i.e., Planetoid) in terms of $F_1$ on BlogCatalog dataset. We observe that Planetoid and MetaTNE achieve comparable performance when $K_{*,+}$ is the same as or larger than $K_{*,-}$, while the performance gap between MetaTNE and Planetoid gradually increases as the ratio of $K_{*,+}$ to $K_{*,-}$ decreases, which demonstrates the practicability of our method since the positive nodes are relatively scarce compared with the negative ones in many realistic applications.

(a) $K_{*,+} = 10$.  (b) $K_{*,+} = 20$.

Figure 3: The performance w.r.t. the numbers of positive and negative nodes on BlogCatalog dataset.

Table 2: Results with standard deviation on few-shot node classification tasks with novel labels. OOM means out of memory (16 GB GPU memory).

(a) $K_{*,+} = 10$ and $K_{*,-} = 20$.

| Method | BlogCatalog | | | Flickr | | |
|---|---|---|---|---|---|---|
| | AUC | $F_1$ | Recall | AUC | $F_1$ | Recall |
| LP | $0.6422^{\pm0.0289}$ | $0.1798^{\pm0.0198}$ | $0.2630^{\pm0.0309}$ | $0.8196^{\pm0.0175}$ | $0.4321^{\pm0.0392}$ | $0.4989^{\pm0.0492}$ |
| LINE | $0.6690^{\pm0.0323}$ | $0.2334^{\pm0.0499}$ | $0.1595^{\pm0.0403}$ | $0.8593^{\pm0.0145}$ | $0.6194^{\pm0.0334}$ | $0.5418^{\pm0.0382}$ |
| Node2vec | $0.6697^{\pm0.0325}$ | $0.3750^{\pm0.0478}$ | $0.2940^{\pm0.0432}$ | $0.8504^{\pm0.0151}$ | $0.6664^{\pm0.0284}$ | $0.6147^{\pm0.0332}$ |
| Planetoid | $0.6850^{\pm0.0320}$ | $0.4657^{\pm0.0437}$ | $0.4301^{\pm0.0451}$ | $\mathbf{0.8601}^{\pm0.0360}$ | $0.6638^{\pm0.0796}$ | $0.6331^{\pm0.0821}$ |
| GCN | $0.6643^{\pm0.0288}$ | $0.3892^{\pm0.0423}$ | $0.3379^{\pm0.0401}$ | OOM | OOM | OOM |
| Meta-GNN | $0.6533^{\pm0.0362}$ | $0.3567^{\pm0.0364}$ | $0.2962^{\pm0.0398}$ | OOM | OOM | OOM |
| MetaTNE | $\mathbf{0.6986}^{\pm0.0305}$ | $\mathbf{0.5380}^{\pm0.0342}$ | $\mathbf{0.6203}^{\pm0.0375}$ | $0.8462^{\pm0.0164}$ | $\mathbf{0.7118}^{\pm0.0223}$ | $\mathbf{0.7700}^{\pm0.0227}$ |
| %Improv. | 1.99 | 15.53 | 44.22 | -1.62 | 6.81 | 21.62 |

| Method | PPI | | | Mashup | | |
|---|---|---|---|---|---|---|
| | AUC | $F_1$ | Recall | AUC | $F_1$ | Recall |
| LP | $0.6285^{\pm0.0221}$ | $0.2147^{\pm0.0384}$ | $0.2769^{\pm0.0630}$ | $0.6488^{\pm0.0258}$ | $0.3103^{\pm0.0414}$ | $0.4535^{\pm0.0991}$ |
| LINE | $0.6372^{\pm0.0270}$ | $0.2147^{\pm0.0373}$ | $0.1456^{\pm0.0280}$ | $0.6926^{\pm0.0354}$ | $0.2970^{\pm0.0602}$ | $0.2142^{\pm0.0537}$ |
| Node2vec | $0.6273^{\pm0.0258}$ | $0.3545^{\pm0.0350}$ | $0.2860^{\pm0.0326}$ | $0.6575^{\pm0.0303}$ | $0.3835^{\pm0.0413}$ | $0.3147^{\pm0.0396}$ |
| Planetoid | $0.6791^{\pm0.0251}$ | $0.4672^{\pm0.0314}$ | $0.4411^{\pm0.0328}$ | $0.7056^{\pm0.0223}$ | $0.4825^{\pm0.0287}$ | $0.4218^{\pm0.0334}$ |
| GCN | $0.6596^{\pm0.0223}$ | $0.4176^{\pm0.0335}$ | $0.3729^{\pm0.0327}$ | $0.6910^{\pm0.0248}$ | $0.4065^{\pm0.0417}$ | $0.3607^{\pm0.0396}$ |
| Meta-GNN | $0.6537^{\pm0.0307}$ | $0.3964^{\pm0.0343}$ | $0.3373^{\pm0.0405}$ | $0.7093^{\pm0.0317}$ | $0.4689^{\pm0.0389}$ | $0.4202^{\pm0.0384}$ |
| MetaTNE | $\mathbf{0.6865}^{\pm0.0205}$ | $\mathbf{0.5188}^{\pm0.0209}$ | $\mathbf{0.5621}^{\pm0.0311}$ | $\mathbf{0.7645}^{\pm0.0251}$ | $\mathbf{0.5764}^{\pm0.0291}$ | $\mathbf{0.5566}^{\pm0.0337}$ |
| %Improv. | 1.09 | 11.04 | 27.43 | 7.78 | 19.46 | 22.73 |

(b) $K_{*,+} = 10$ and $K_{*,-} = 40$.

| Method | BlogCatalog | | | Flickr | | |
|---|---|---|---|---|---|---|
| | AUC | $F_1$ | Recall | AUC | $F_1$ | Recall |
| LP | $0.6421^{\pm0.0288}$ | $0.0554^{\pm0.0118}$ | $0.0727^{\pm0.0158}$ | $0.8253^{\pm0.0156}$ | $0.3055^{\pm0.0413}$ | $0.3040^{\pm0.0485}$ |
| LINE | $0.6793^{\pm0.0320}$ | $0.0529^{\pm0.0316}$ | $0.0328^{\pm0.0216}$ | $0.8644^{\pm0.0139}$ | $0.4154^{\pm0.0471}$ | $0.3485^{\pm0.0471}$ |
| Node2vec | $0.6792^{\pm0.0314}$ | $0.1982^{\pm0.0516}$ | $0.1340^{\pm0.0398}$ | $0.8558^{\pm0.0150}$ | $0.5295^{\pm0.0381}$ | $0.4602^{\pm0.0420}$ |
| Planetoid | $0.6981^{\pm0.0315}$ | $0.2980^{\pm0.0550}$ | $0.2319^{\pm0.0507}$ | $\mathbf{0.8728}^{\pm0.0382}$ | $0.5040^{\pm0.0790}$ | $0.4461^{\pm0.0741}$ |
| GCN | $0.6794^{\pm0.0302}$ | $0.2104^{\pm0.0347}$ | $0.1583^{\pm0.0268}$ | OOM | OOM | OOM |
| Meta-GNN | $0.6724^{\pm0.0396}$ | $0.2152^{\pm0.0578}$ | $0.1618^{\pm0.0546}$ | OOM | OOM | OOM |
| MetaTNE | $\mathbf{0.7139}^{\pm0.0309}$ | $\mathbf{0.4398}^{\pm0.0401}$ | $\mathbf{0.5819}^{\pm0.0451}$ | $0.8505^{\pm0.0154}$ | $\mathbf{0.6220}^{\pm0.0245}$ | $\mathbf{0.7460}^{\pm0.0523}$ |
| %Improv. | 2.26 | 47.58 | 150.93 | -2.55 | 17.47 | 62.10 |

| Method | PPI | | | Mashup | | |
|---|---|---|---|---|---|---|
| | AUC | $F_1$ | Recall | AUC | $F_1$ | Recall |
| LP | $0.6298^{\pm0.0228}$ | $0.0773^{\pm0.0231}$ | $0.0748^{\pm0.0277}$ | $0.6534^{\pm0.0259}$ | $0.1156^{\pm0.0276}$ | $0.1284^{\pm0.0509}$ |
| LINE | $0.6423^{\pm0.0268}$ | $0.0496^{\pm0.0193}$ | $0.0300^{\pm0.0122}$ | $0.7009^{\pm0.0345}$ | $0.0956^{\pm0.0489}$ | $0.0617^{\pm0.0348}$ |
| Node2vec | $0.6309^{\pm0.0264}$ | $0.1894^{\pm0.0373}$ | $0.1306^{\pm0.0286}$ | $0.6643^{\pm0.0311}$ | $0.2070^{\pm0.0417}$ | $0.1447^{\pm0.0333}$ |
| Planetoid | $0.6879^{\pm0.0250}$ | $0.3100^{\pm0.0368}$ | $0.2523^{\pm0.0323}$ | $0.7095^{\pm0.0223}$ | $0.3279^{\pm0.0298}$ | $0.2551^{\pm0.0278}$ |
| GCN | $0.6608^{\pm0.0225}$ | $0.2531^{\pm0.0353}$ | $0.1974^{\pm0.0268}$ | $0.7007^{\pm0.0245}$ | $0.2558^{\pm0.0237}$ | $0.2098^{\pm0.0169}$ |
| Meta-GNN | $0.6617^{\pm0.0309}$ | $0.2575^{\pm0.0332}$ | $0.2088^{\pm0.0396}$ | $0.7140^{\pm0.0339}$ | $0.3412^{\pm0.0554}$ | $0.2864^{\pm0.0635}$ |
| MetaTNE | $\mathbf{0.7039}^{\pm0.0218}$ | $\mathbf{0.4298}^{\pm0.0242}$ | $\mathbf{0.5327}^{\pm0.0420}$ | $\mathbf{0.7684}^{\pm0.0244}$ | $\mathbf{0.4814}^{\pm0.0318}$ | $\mathbf{0.4816}^{\pm0.0393}$ |
| %Improv. | 2.33 | 38.65 | 111.14 | 7.62 | 41.09 | 68.16 |

## 3.3 The Performance w.r.t. the Number of Query Nodes

In the above experiments, we presume that, for each few-shot node classification task, the support and the query sets have the same numbers of positive and negative nodes following the standard protocol of meta-learning (called the *standard-setting*). However, in practice, the query set could have different numbers of positive and negative nodes as well as a different ratio of the number of positive nodes to the number of negative nodes compared to the support set. Thus, we further examine how the number of query nodes influences the performance. Towards this end, we sample additional test

(a) $K_{\mathcal{Q},+}^{\text{test}} = 10$.    (b) $K_{\mathcal{Q},+}^{\text{test}} = 20$.

Figure 4: The performance w.r.t. the number of query nodes on PPI dataset.

tasks by varying the numbers of positive and negative nodes in the query set (i.e., $K_{\mathcal{Q},+}^{\text{test}}$ and $K_{\mathcal{Q},-}^{\text{test}}$), with the numbers of positive and negative nodes in the support set fixed at 10 and 30, respectively (i.e., $K_{\mathcal{S},+}^{\text{test}} = 10$ and $K_{\mathcal{S},-}^{\text{test}} = 30$), and then compare the performance on these tasks. This setting is called the *generalized-setting*. Note that here we only alter the sampling of test tasks as described above and the training tasks are always sampled under the condition that both the support and query sets contain 10 positive and 30 negative nodes (i.e., $K_{*,+}^{\text{train}} = 10$ and $K_{*,-}^{\text{train}} = 30$). Figure 4 shows the experimental results on PPI dataset.

We observe that MetaTNE consistently yields better performance than Planetoid under different combinations of $K_{\mathcal{Q},+}^{\text{test}}$ and $K_{\mathcal{Q},-}^{\text{test}}$. In particular, jointly analyzing Table 2 and Fig. 4a, MetaTNE achieves almost the same performance in both the standard- and generalized-settings when the query set contains 10 positive nodes as well as 20 or 40 negative nodes, which indicates that to some extent MetaTNE is not sensitive to the choice of $K_{*,+}$ and $K_{*,-}$ for sampling training tasks as well as $K_{\mathcal{S},+}^{\text{test}}$ and $K_{\mathcal{S},+}^{\text{test}}$ and demonstrates the robustness of MetaTNE. On the other hand, it essentially becomes easier to classify the query nodes as the ratio of $K_{\mathcal{Q},+}^{\text{test}}$ to $K_{\mathcal{Q},-}^{\text{test}}$ increases, whereas the performance of Planetoid does not change markedly as $K_{\mathcal{Q},-}^{\text{test}}$ decreases in Fig. 4, which evidences that Planetoid tends to overfit the training tasks (e.g., the ratio of the number of positive nodes to the number of negative nodes).

### 3.4 The Performance with Fewer Positive Nodes

We further examine the performance of different methods by using fewer positive nodes and conduct experiments with $K_{*,+}$ set to 5 and $K_{*,-}$ set to 10 or 20. Table 3 reports the experimental results on BlogCatalog dataset. From Table 3, we observe similar results to Table 2 and MetaTNE still significantly outperforms all other methods in the case that there are fewer positive nodes.

Table 3: Results of fewer positive nodes on BlogCatalog dataset.

| Method | $K_{*,+} = 5, K_{*,-} = 10$ | | | $K_{*,+} = 5, K_{*,-} = 20$ | | |
|---|---|---|---|---|---|---|
| | AUC | F$_1$ | Recall | AUC | F$_1$ | Recall |
| LP | $0.6231^{\pm 0.0284}$ | $0.1753^{\pm 0.0168}$ | $0.2831^{\pm 0.0279}$ | $0.6226^{\pm 0.0288}$ | $0.0567^{\pm 0.0101}$ | $0.0930^{\pm 0.0159}$ |
| LINE | $0.6355^{\pm 0.0295}$ | $0.1296^{\pm 0.0379}$ | $0.0884^{\pm 0.0291}$ | $0.6432^{\pm 0.0300}$ | $0.0116^{\pm 0.0141}$ | $0.0076^{\pm 0.0098}$ |
| Node2vec | $0.6384^{\pm 0.0299}$ | $0.2912^{\pm 0.0440}$ | $0.2267^{\pm 0.0387}$ | $0.6451^{\pm 0.0305}$ | $0.1017^{\pm 0.0372}$ | $0.0689^{\pm 0.0273}$ |
| Planetoid | $0.6473^{\pm 0.0303}$ | $0.4221^{\pm 0.0408}$ | $0.4052^{\pm 0.0437}$ | $0.6583^{\pm 0.0318}$ | $0.2305^{\pm 0.0509}$ | $0.1853^{\pm 0.0470}$ |
| GCN | $0.6379^{\pm 0.0308}$ | $0.3376^{\pm 0.0473}$ | $0.3015^{\pm 0.0455}$ | $0.6524^{\pm 0.0312}$ | $0.1590^{\pm 0.0492}$ | $0.1239^{\pm 0.0408}$ |
| Meta-GNN | $0.6392^{\pm 0.0362}$ | $0.3523^{\pm 0.0375}$ | $0.3152^{\pm 0.0468}$ | $0.6552^{\pm 0.0399}$ | $0.1719^{\pm 0.0612}$ | $0.1485^{\pm 0.0598}$ |
| MetaTNE | $\mathbf{0.6546}^{\pm 0.0286}$ | $\mathbf{0.4523}^{\pm 0.0371}$ | $\mathbf{0.4842}^{\pm 0.0469}$ | $\mathbf{0.6756}^{\pm 0.0295}$ | $\mathbf{0.3730}^{\pm 0.0387}$ | $\mathbf{0.4539}^{\pm 0.0505}$ |
| %Improv. | 1.13 | 7.15 | 19.50 | 2.63 | 61.82 | 144.95 |

### 3.5 Visualization

To better demonstrate the effectiveness of the transformation function, we select two typical query nodes from the test tasks on Flickr dataset and visualize the relevant node embeddings before and after adaptation with t-SNE [7] in Fig. 5. Note that "Query (+)" and "Query (-)", respectively, indicate

the adapted embeddings of the query node in relation to the positive and negative support nodes in Eqn. (2). From Fig. 5a where the label of the query node is negative, we see that, before adaptation, the embedding of the query node is closer to the positive prototype than the negative prototype and thus misclassification occurs. After adaptation, the distance between "Query (-)" and the negative prototype is smaller than that between "Query (+)" and the positive prototype and hence the query node is classified correctly. The similar behavior is observed in Fig. 5b. Moreover, we observe that the transformation function is capable of either (1) gathering the positive and negative support nodes into two separate regions as shown in Fig. 5a or (2) adjusting "Query (+)" and "Query (-)" to make the right prediction when the positive and negative prototypes are close as shown in Fig. 5b. Another observation is that the transformation function has the tendency of enlarging the distances between node embeddings to facilitate classification.

(a) The ground-truth of the query node is negative.

(b) The ground-truth of the query node is positive.

Figure 5: t-SNE visualization of embedding adaptation.

## Footnotes

[1] http://socialcomputing.asu.edu/datasets/BlogCatalog3

[2] http://socialcomputing.asu.edu/datasets/Flickr

[3] https://snap.stanford.edu/node2vec/

[4] https://github.com/xiangyue9607/BioNEV

[5] `https://github.com/yamaguchiyuto/label_propagation`

[6] Since the authors do not provide the implementation that uses GCN as the learner, we implement it on the basis of the released code at `https://github.com/ChengtaiCao/Meta-GNN` to perform experiments.