[Reviews · NeurIPS 2020]

Review 1

Summary and Contributions: The paper proposes a method that tackles the problem of learning graph node classification of tasks that have few labelled samples. The Node Classification on graphs with Few-shot Novel Labels (NCFNL) assumes that we have many labelled samples from a set of known labels but we want to learn tasks that have few labelled samples from novel classes. The paper adopts the meta-learning framework used in few-shot image classification and builds upon methods from that domain. The core part of the method is adapted from image classification and uses the Prototype Networks to predict the probability of novel classes based on distances to a support set of few labelled examples. The features of the graph nodes are learned with a skip-gram metho. The method adapts the query and prototype features using self-attention and suggests that forming separate embeddings for positive and negative samples is beneficial. A training scheduler is proposed where optimisation of the embedding by the skip-gram method is alternated with the learning of the few shot objective in

Strengths: The problem of learning from a few new labels for graph data is important and following the meta-learning framework established in image classification is a good direction. The usage of a prototype network in the context of graph data is sound. The proposed self-attention for adapting the samples to the tasks given by the support set is well motivated and makes sense. The separate attention transformation and the training procedure are experimentally shown to produce performance improvements. The ablation studies are appreciated. The paper is mainly clearly written and well-motivated.

Weaknesses: The main weakness of the paper is the evaluation of the baselines. The current method uses a skip-gram method to learn node embeddings from the structure of the graph, without using any existing node representation. The skip-gram model is similar to node2vec and this comparison is fair. But GCN and Meta-GNN are designed to use nodes with features. In [Kipf and Welling 2017], GCN is used with one-hot features only for synthetic, random graph experiments. A fair comparison would be to use GCN and Meta-GNN with node2vec embeddings or with the skip-gram model that is pre-trained. There could be more ablation studies. What happens if the node embeddings are learned at the beginning by the Structural Module and then left fixed? What is the performance using just one-hot features? Could the authors comment why in Figure 3 from the Supplement the performance is degraded when more negative samples are given? Shouldn’t the performance increase or at least remain constant? More explanations would be useful. =================== Post-Rebuttal: =================== I thank the authors for the additional explanations and experiments in the rebuttal. My main concerns were regarding the comparison with the GNN baselines having proper node features and the authors mainly addressed my concerns in the experiments from the rebuttal. Although comparisons with the baselines having the exact input features in the Meta-learning Module would have been also appreciated. I mainly agree with R3 that the paper has somehow limited contributions. But on the other hand, the direction of using few-samples on graph-based data is relatively new with few existing papers and this work could further increase the interest in this domain. Together with Meta-GNN, this work could be a good baseline for future works in few-shot for graph data. Thus I am increasing my score to a 6.

Correctness: The claims and the method are sound.

Clarity: The paper is mainly well written.

Relation to Prior Work: The setting of few-shot classification is not yet well explored for graph data. The paper discusses existing work on graph data, but the evaluation of previous methods is not entirely fair.

Reproducibility: Yes

Additional Feedback: Is there a reason why the proposed method shouldn’t be used for tasks where node features exist?


Review 2

Summary and Contributions: The authors studied few-shot learning for node classification. More specifically, they named the problem as Node Classification on graphs with Few-shot Novel Labels (NCFNL). This can address cool start problems in recommender systems and classifications for new issues/contents on social media. The proposed method consists of three modules: structural module to learn task-agnostic node embedding, a meta-learning module to transform the task-agnostic embeddings to task-specific embeddings, and optimization module to balance the graph structure learning (task-agnostic learning) and transformation learning/meta-learning (task-specific learning). The experimental results look promising.

Strengths: 1. Strong experimental results. The proposed method shows strong experimental results. It outperforms strong baselines by a significant margin. The largest cases are 150% and 47.58% in terms of recall and F1. 2. The problem definition of Node Classification on Graphs With Few-Shot Novel Labels is interesting. It is relevant to semi-supervised learning, continual learning, and few-shot learning. The NCFNL subsumes many problems in a variety of applications such as recommender systems for new items with a few ratings and classification for new labels. The graphs can be constructed using the proximity of samples. So, it has a wide operating range.

Weaknesses: 1. The meta-learning formulation is too brief. It was not clear how the support, query sets are used in Eq. (6). 2. The dimensionality of variables (e.g., X, W) should be added X in R^{d1xd2} as authors did in the supplement sec 1.1. 3. The authors may want to shorten section 4.2.2. and its wordy discussion about embedding transformation to save more space for meta-learning formulation. 4. I doubt that the optimization part deserves the "module". Authors provide a way to balance graph structural learning and meta-learning. But it does not come with theoretical or empirical justification.

Correctness: Yes, the claims and prosed method are correct.

Clarity: Yes, overall the manuscript is well-written. Due to the limited space, some important parts seem too shortened as discussed in the weakensses. Other than that, the manuscript reads well.

Relation to Prior Work: Yes, it discussed the difference between the proposed method and few-shot learning on graphs and why the prior works are not directly applicable to the problem in this paper.

Reproducibility: Yes

Additional Feedback: =================== Post-Rebuttal: =================== I have read the detailed author feedback. Overall, this submission studies an interesting problem and formulation. It can be used for few-shot classification and cool-start recommendation. Many meta-learning papers are similar to MAML in the way to handle bi-level optimization. But as the authors said they provided a transformation function and a training scheduler. The technical contributions are somewhat limited but this paper will be a great baseline for few-shot node classification methods later.


Review 3

Summary and Contributions: This paper introduces node classification on graphs with few-shot novel labels, and proposes a model called MetaTNE to handle this problem. MetaTNE has three components: 1) the structural module, which utilizes a traditional graph embedding approach for node embedding; 2) the meta-learning module, which employs the prototypical meta-learning based model to capture the relationships between the graph structure and the node labels, and an embedding transformation function for embedding adaptation; 3) the optimization module to optimize the whole loss iteratively. The research direction is interesting and hot, and the structure of the paper is well-organized. However, there are still some weaknesses that should be taken into consideration: 1) the contribution is limited in some sense; 2) some explanations are needed to clarify some claims; 3) some more baselines should be considered. *** Update **** I read the authors' response. It well addressed some of my questions in experimental study and the use of self attention. However, the overall novelty of this work is still limited; the proposed methodology is somewhat incremental building upon several well known concepts such as graph structural learning, attention and meta-learning.

Strengths: 1. The research direction is interesting and hot. 2. The structure of the paper is well-organized.

Weaknesses: 1. The overall novelty of the proposed model is limited to some extend. 1) The structural module is generally a traditional graph embedding approaches; 2) the meta-learning module utilizes the prototypical model to calculate the centers of the positive and negative nodes respectively, and the transformation generally leverages an self-attention mechanism to achieve the adapted node embeddings for different labels. I think this module is very similar to meta-GNN. Both of them conduct adaptation on the support set, then do evaluation on the query set, though they employ prototype and MAML respectively. 3) the optimization module applies a simple iteration strategy to optimize the two loss functions. In my view, the overall model stands on the shoulder on some traditional approaches, and seems a bit incremental. 2. For the motivation, why to use meta-learning? Could some other approaches, such as fine-tune (which is often utilized as the comparison with meta-learning), solve this novel label problem? The authors should give more explanations, and verify them by experiments. 3. Some concerns about the first contribution "To the best of our knowledge, this is the first work that only uses the graph structure and some known labels to study the problem of NCFNL". I think this contribution is over claimed. Actually, Meta-GNN [33] also utilizes the graph structure (GNN) and some known labels (the labeled nodes) to study the problem of NCFNL (to predict the node labels for novel classes). We cannot regard it as overlooking the graph structure just because it utilizes the GNN models but not graph embedding models. 4. I agree with the usage of transformation, which actually does adaptation to transform the task agnostic embeddings to some task-specific ones for each task. My concern is that, why to use self-attention as the transformation, and what is the insight? The authors should give more explanations. 5. The baselines are not quite sufficient. 1) For GNN models, the authors could apply the graph embeddings learned by some traditional approaches (e.g., deepwalk) as the node features, which would be better than identity matrix, since the graph embeddings could preserve the graph structure. 2) Some more baselines should be taken into consideration, such as the fine-tuning approaches. For example, we can first train a GCN model on the training data for label prediction, then in test we can fine tune the GNN parameters on the novel labels for label prediction. I think fine-tuning approaches are important baselines for meta-learning models, and this could verify the performance comparison between fine-tuning methods and meta-learning based approaches.

Correctness: Generally correct.

Clarity: Generally good.

Relation to Prior Work: More discussions are needed to clarify the differences between this work and some previous related studies.

Reproducibility: Yes

Additional Feedback: Please see Weaknesses.


Review 4

Summary and Contributions: The authors focused on the few-shot node classification problem and proposed a novel framework/model, metaTNE, which learns a generalized node classifier from sufficient known label data to classify nodes for novel labels. The intuition behind the proposed metaTNE is that different labels may have similar propagation pattern, if we can learn the pattern from existing labels data, we may transfer this knowledge to other kinds of labels. The proposed metaTNE sounds, and can potentially solve many real-world problems, e.g., recommendations for newly formed groups with only a few users in online social networks. The authors conducted extensive experiments, where the results demonstrated the effectiveness of the proposed metaTNE. Feedback to the rebuttal Update: The response was well received. Overall, the authors addressed a crucial problem, but the technical contribution seems limited. I will keep the review rating.

Strengths: 1. This paper is well-organized, the idea is novel to solve the node classification problem on graphs in a meta-learning framework. 2. The proposed metaTNE sounds, which is composed of three modules: structural module, meta-learning module and optimization module. structural module is used for node embedding, meta-learning module uses the metric-based algorithm to meta-learn the classifier, and optimization module is used to optimize the previous two modules. The three modules can effectively solve the node classification problem. 3. Extensive experiments on four different popular graph datasets were conducted, the experimental results are reasonable compared with state-of-the-art baseline models.

Weaknesses: W1. In the problem definition, the authors define the label indicator for all nodes as 0 or 1 indicating a node holding or not holding a label. However, in many cases, the labels may be uncertain/probabilistic. It could be more realistic and interesting to model such uncertainty in the proposed model. W2. In addition, in structural model, the authors use 1-hop neighbors to construct the neighboring net, it would be nice to provide more details or conduct experiments to explain why using 1-hop neighbors is sufficient. W3. It is not clear why self-attention is a good choice to learn the task-specific representation. The authors could show some ablation study over self-attention vs other alternative models.

Correctness: Yes, the claims and the methods are correct and the empirical methodology is also correct.

Clarity: Yes. This paper is well presented, the organization is clear.

Relation to Prior Work: Yes. The authors provided detailed info of the related works. The differences between this work and previous works are also explained.

Reproducibility: Yes

Additional Feedback:

[Author Response · NeurIPS 2020]

We would like to thank each of the reviewers for the constructive and insightful comments on our manuscript.

**R1, R3: Comparison with more baselines.** Table 1 shows more
comparison results, where the suffixes "-I" and "-P", respectively,
indicate that the identity matrix and the pretrained node2vec embed-
dings are used as the input features. We observe that the pretrained
structural embeddings can indeed bring performance improvement.
However, our MetaTNE still outperforms GCN-P and Meta-GNN-
P by a significant margin. In addition, we see that Meta-GNN-P
underperforms GCN-P and the reason is discussed in lines 311-315
in our paper. Due to limited space here, we will give the results of GCN-P and Meta-GNN-P on other datasets as well
as with different $K_{*,+}$ and $K_{*,-}$, in the final version.

Table 1: Results with $K_{*,+} = 10$ and $K_{*,-} = 20$.

| Method | BlogCatalog | | | PPI | | |
|---|---|---|---|---|---|---|
| | AUC | $F_1$ | Recall | AUC | $F_1$ | Recall |
| GCN-I | 0.6102 | 0.2730 | 0.2194 | 0.6544 | 0.3379 | 0.2721 |
| GCN-P | 0.6643 | 0.3892 | 0.3379 | 0.6596 | 0.4176 | 0.3729 |
| Meta-GNN-I | 0.4805 | 0.2375 | 0.2141 | 0.5466 | 0.3289 | 0.3081 |
| Meta-GNN-P | 0.6533 | 0.3567 | 0.2962 | 0.6537 | 0.3964 | 0.3373 |
| MetaTNE | **0.6986** | **0.5380** | **0.6203** | **0.6865** | **0.5188** | **0.5621** |

**R1: (1) More ablation studies.** Table 2 gives more ablation study
results, where V1 denotes that the node embeddings are learned at
the beginning and then left fixed and V2 denotes that each node
is represented by a one-hot vector. Our method significantly out-
performs V1 and V2 and the performance of these two variants is
worse than that of the variants in Table 3 in our paper. In addition,
V1 underperforms V2 even if the node embeddings of V1 are first

Table 2: Results of ablation study in terms of $F_1$.

| Method | $K_{*,+} = 10, K_{*,-} = 20$ | | $K_{*,+} = 10, K_{*,-} = 40$ | |
|---|---|---|---|---|
| | BlogCatalog | PPI | BlogCatalog | PPI |
| MetaTNE | **0.5380** | **0.5188** | **0.4398** | **0.4298** |
| V1 | 0.4748 | 0.4614 | 0.3549 | 0.3389 |
| V2 | 0.4892 | 0.4819 | 0.3699 | 0.3777 |

learned from the graph structure. We speculate that the reason is that the latent space of node embeddings somewhat
overfits to the metric of graph structure learning, making it harder to adapt to the metric of subsequent meta-learning or
few-shot learning tasks. **(2) Explanation on Figure 3 in the supplement.** When there are more negative samples and
the number of positive samples is fixed, the data becomes more skewed. A large degree of imbalance leads the classifier
to bias towards the negative samples, which has two impacts: very few samples are predicted as positive samples, and the
true positive samples are more difficult to identify. In general, the recall scores will drop significantly while the precision
scores will not change too much. Consequently, both our method and the baseline show performance degradation in the
$F_1$ scores when more negative samples are given and the number of positive samples keeps unchanged.

**R2: (1) About the meta-learning formulation.** Due to limited space, we place the detailed meta-learning formulation
of how to use the support and query sets in the supplement. We will clarify it in the final version and ensure that the
paper is self-contained. Also, we will further polish our paper based on your suggestions to address other writing issues.
**(2) Empirical justification for the optimization part.** We refer the reviewer to Table 3 in our paper where the results
of V3 empirically justify the effectiveness of the optimization part. **(3) About tasks where node features exist.** Since
we focus more on the featureless scenarios, MetaTNE currently cannot handle node features, and further research is
needed to incorporate node features into the structural and meta-learning modules of our method.

**R3: (1) Comparison against Meta-GNN.** It is a standard paradigm, that both Meta-GNN and our MetaTNE follow, to
conduct adaptation on the support set and then do evaluation on the query set in the meta-learning literature, however, it
is non-trivial to effectively apply meta-learning to the considered *content-less graph data* under the *multi-label setting*.
Our main technical contributions are the *specially designed transformation function and training scheduler*, which
enable MetaTNE to achieve strong experimental results. In contrast, Meta-GNN simply uses MAML to train GCN
models and does not show satisfactory performance in the scenario of interest even if using the node2vec embeddings
as input as shown in Table 1. The reasons are discussed in lines 308-315 in our paper. **(2) Regarding the fine-tuning
approaches.** As mentioned in Sec. 2.2 of the supplement, the baseline GCN is actually evaluated in a fine-tuning
manner. Specifically, we first train a GCN model on the training data and then fine-tune the parameters of the last layer
on the novel labels. During the rebuttal period, we further try fine-tuning all layers on the novel labels and find that
the performance of fine-tuning all layers is slightly worse than that of only fine-tuning the last layer. For example, on
BlogCatalog dataset with $K_{*,+} = 10$ and $K_{*,-} = 20$, the $F_1$ of the former is 0.3746 and the $F_1$ of the latter is 0.3892
(note that these numbers are obtained by using the node2vec embeddings as input). Complete results will be available
in the final version and omitted here due to limited space. Overall, our proposed MetaTNE significantly outperforms
the fine-tuning approaches. **(3)** We will resummarize the first contribution in lines 66-69 to make it more appropriate.

**R3, R5: Explanation on why to use self-attention.** For the embedding transformation, our goal is to find how a query
node correlates with positive or negative support nodes. The self-attention has shown its power to effectively capture
relationships between a set of elements in a wide range of applications and naturally meets our needs. We agree that it
is insightful to explore different architectures to implement the transformation. We have already started working on this
and will report our findings in the final version.

**R5:** By using 1-hop neighbors, our method already outperforms the baselines by a significant margin, and thus we did
not try other ways to construct the neighbor set. In addition, we agree that it is more realistic to model label uncertainty.
We leave these explorations as important future work.

[Meta-Review · NeurIPS 2020]

The paper studies an interesting problem and formulation which can be used for a few-shot classification and cool-start recommendation. The authors provided a novel transformation function and a training scheduler in the MAML framework which the reviewers appreciated. There was some concerns initially but the rebuttal did clarify some of the confusion. In the end, reviewers were convinced that paper offers some novel ideas and it will be a great baseline for few-shot node classification methods later.